# Development of ^99m^Tc-Hynic-Adh-1 Molecular Probe Specifically Targeting N-Cadherin and Its Preliminary Experimental Study in Monitoring Drug Resistance of Non-Small-Cell Lung Cancer

**DOI:** 10.3390/cancers15030755

**Published:** 2023-01-26

**Authors:** Qianni Ye, Zhenfeng Liu, Shuyi Zhang, Guolin Wang, Guanghua Wen, Mengjie Dong

**Affiliations:** 1Department of Radiology, The First Affiliated Hospital of Wenzhou Medical University, Wenzhou 325015, China; 2Department of Nuclear Medicine, The First Affiliated Hospital, Zhejiang University School of Medicine, Hangzhou 310003, China; 3Department of Nuclear Medicine, Shenzhen Longhua District Central Hospital, Shenzhen 518110, China; 4Department of Nuclear Medicine, Peking University Shenzhen Hospital, Shenzhen 518110, China

**Keywords:** tumor resistance, EMT, ADH-1, SPECT/CT imaging, molecular probe

## Abstract

**Simple Summary:**

Non-small-cell lung cancer (NSCLC) represents approximately 80–85% of all lung cancers, and tumor resistance remains common and difficult to treat. Therefore, early detection of tumor resistance is of great significance for improving prognosis. Here, we developed a novel molecular imaging agent that could be used for imaging drug-resistant NSCLC, providing a noninvasive method for dynamically observing whether tumor resistance occurs during treatment.

**Abstract:**

Background: N-cadherin is considered a characteristic protein of EMT and has been found to be closely related to tumor resistance. In this study, a novel molecular imaging probe, ^99m^Tc-HYNIC-ADH-1, was developed, and its diagnostic value in monitoring drug resistance in NSCLC was preliminarily investigated. Methods: ADH-1 was labeled indirectly with ^99m^Tc. Radiochemical purity and stability, partition coefficients and pharmacokinetics were evaluated. Additionally, the fluorescent probe of ADH-1 was synthesized to study tumor uptake in cells level and in vivo. Biodistribution analysis and small animal SPECT/CT were performed in PC9GR and PC9 tumor-bearing mice. Results: ^99m^Tc-HYNIC-ADH-1 was highly stable (radiochemical purity ≥ 98% in PBS and serum after 24 h). A cell binding study and fluorescence imaging showed that the uptake was significantly higher in PC9GR cells (gefitinib-resistant) than in PC9 cells (nonresistant) (*p* < 0.05). Biodistribution analysis showed rapid blood clearance and significant uptake in the kidney and resistant tumor. Small animal SPECT/CT studies showed that uptake in PC9GR tumors (T/NT = 7.73 ± 0.54) was significantly higher than that in PC9 tumors (T/NT = 3.66 ± 0.78) at 1 h (*p* = 0.002). Conclusions: The ^99m^Tc-HYNIC-ADH-1 molecular probe has a short synthesis time, high labeling rate, high radiochemical purity and good stability, does not require purification, is characterized by rapid blood clearance and is mainly excreted through the urinary system. ^99m^Tc-HYNIC-ADH-1 is considered a promising probe for monitoring drug resistance in NSCLC.

## 1. Introduction

Lung cancer is one of the most common and deadly malignant tumors worldwide, accounting for the largest number of cancer-related deaths. Approximately 80–85% of lung cancers are non-small-cell lung cancer (NSCLC) [1]. In the treatment of lung cancer, tyrosine kinase inhibitors (TKIs) targeting epidermal growth factor receptor (EGFR) have been widely used. For patients with EGFR gene mutations, the effective rate is 60–70% [2,3]. However, with the extension of treatment, most originally sensitive patients gradually develop acquired drug resistance, which eventually leads to poor prognosis [4,5].

Epithelial–mesenchymal transition (EMT) refers to the process of transforming epithelial phenotype cells into mesenchymal phenotype cells, which plays an indispensable physiological role in embryonic development, organ formation, damage repair and so on [6]. In pathological EMT, recent new evidence has shown that it is involved in the process of acquired drug resistance in tumorigenesis [7]. EMT is usually characterized by decreased intercellular adhesion, loss of polarity and formation of pseudopodia. Immunohistochemistry showed that the epithelial signature protein (E-cadherin) was downregulated, and the interstitial signature proteins, such as N-cadherin and Vimentin, were upregulated [8]. Many studies have found that the phenotype of cells resistant to EGFR-TKIs is consistent with that of EMT [9,10]. Lee et al. showed that, compared with their parental strains, the gefitinib-resistant strains PC9 and HCC827 showed significantly upregulated expression of interstitial signature proteins, without the T790M mutation and MET amplification, showing an EMT phenotype. Moreover, blocking the EMT signaling pathway can reverse EMT and restore sensitivity to gefitinib [11]. Therefore, EMT characteristic proteins such as N-cadherin may be important targets for reversing EMT and restoring tumor sensitivity.

N-cadherin is a transmembrane glycoprotein that mediates intercellular adhesion, cell migration and blood vessel formation and stability. Overexpression of N-cadherin often leads to decreased adhesion, loss of polarity and resistance to apoptosis in tumor cells, and N-cadherin is considered one of the interstitial markers of EMT [12]. N-cadherin has been proven to be overexpressed in a variety of tumor types. In NSCLC, the high expression of N-cadherin is not only related to tumor stage, differentiation degree and clinical prognosis, but is also closely related to gefitinib resistance and lung cancer brain metastasis [13,14,15]. Given the important role of N-cadherin in tumorigenesis, tumor migration and drug resistance, it is of great significance to study N-cadherin inhibitors [16]. Several inhibitors of N-cadherin, including synthetic linear peptides, synthetic cyclic peptides, non-peptide pepsin inhibitors and monoclonal antibodies, have been identified. ADH-1 (N-Ac-CHAVC-NH2) is a synthetic cyclic peptide, the most-studied inhibitor and the only N-cadherin inhibitor to enter clinical trials [17,18,19]. ADH-1 has been shown to induce the apoptosis of various tumor cells, inhibit tumor cell migration and improve drug sensitivity in both in vitro and preclinical experiments [20,21,22].

In tumor treatment, the development of molecular probes with high specificity and high sensitivity for the early detection of tumor drug resistance is of great significance for the timely adjustment of personalized treatment and improvement of prognosis [23]. As a specific inhibitor of N-cadherin, an EMT mesenchymal marker, ADH-1 has been confirmed to have high antitumor activity and mild side effects, presenting the essential features for promising molecular probes.

In this study, N-cadherin was used as the biological target. First, ADH-1 fluorescent molecular probe was synthesized to explore its targeting of or specificity for resistant tumor cells in vitro and in vivo. Hydrazine nicotinamide (HYNIC) was used as the chelating agent to synthesize the coupling compound of precursor HYNIC-ADH-1, and ^99m^Tc-HYNIC-ADH-1, a single-photon molecular probe, was synthesized through radiochemistry with ^99m^Tc as a single-photon nuclide. Small animal SPECT imaging was performed in tumor-bearing mice, laying a solid foundation for further exploration of tumor resistance.

## 2. Materials and Methods

### 2.1. Materials

Cy3-ADH-1 (Appendix A) and Cy7-ADH-1 (Appendix A) were synthesized by Tanzhen Biotechnologies Co., Ltd. (Nanchang, China). ADH-1 (Appendix A) and HYNIC-ADH-1 (Appendix A) were synthesized by Peptites Biotech Co., Ltd. (Zhejiang, China). ^99m^TcO_4_Na was provided by Atomic High-tech Pharmaceutical Co., Ltd. (Hangzhou, China). Tricine and Ethylenediamine diacetic acid (EDDA) were purchased from Sigma-Aldrich (Saint Louis, MO, USA). Stannous chloride dehydrate (SnCl_2_) was purchased from Guanghua Sci-tech Co., Ltd. (Guangdong, China). Unless stated otherwise, all other chemicals were obtained from Biological Industries (Kibbutz Beit-Haemek, Israel). A constant-temperature oil bath was purchased from IKA (Staufen, Germany). The Gamma-counting instrument was purchased from Rihuan Instrument (Shanghai, China). HPLC was purchased from Agilent Technologies (Santa Clara, CA, USA). Products were isolated and purified using an HPLC C18 column (Zorbax SB-C18 4.6 × 250 mm, 5 μm). HPLC conditions were acetonitrile solution (dissolved in 0.1% trifluoroacetic acid) as the mobile phase at a flow rate of 1.0 mL/min.

### 2.2. Cell Culture

NSCLC cell lines, PC9 and its gefitinib-resistant strain PC9GR were obtained from the China Infrastructure of Cell Line Resources, and were cultured in DMEM medium containing 10% fetal bovine serum (FBS) in a humidified atmosphere of 5%CO_2_ at 37 °C. The PC9GR cell line maintained its resistance at 1 μg/mL gefitinib.

### 2.3. Animals and Xenograft Models

All animal research was performed in accordance with the Zhejiang University Institutional Animal Care and Use Committee guidelines. PC9 and PC9GR cells (2 × 10^6^ cells per mice) were injected subcutaneously into the flanks of 2- to 4-week old BALB/c nude mice (left: PC9GR, right: PC9). The cells were allowed to grow 3–5 weeks until tumors were 1 ± 0.5 cm^3^ in volume. Normal C57/BL6 mice (aged 6 to 8 wks old) were obtained from the Animal Center of the First Affiliated Hospital of Zhejiang University. All animals were maintained with free access to food and water and used a standard diet, bedding and environment.

### 2.4. Radiochemistry and Different Influencing Factors on Labeling Yield

In different reaction conditions, the effects of reaction time (1, 5, 10, 20, 30 and 60 min), tricine (1, 2, 5, 10, 15 and 20 mg), HYNIC-ADH-1 (10, 20, 30, 40, 50, 100 and 200 μg) and SnCl_2_ (0, 10, 20, 30 and 50 μg) dosage on the labeling yield were investigated. The labeling rate of ^99m^Tc-HYNIC-ADH-1 was measured using the ITLC-SG. Acetone and acetonitrile aqueous solution (V:V = 1:1) were used as the expanding agent, and the solution was expanded to the top in a closed container. After removal, the solution was dried and divided into 10 parts. The counting of each part was measured and recorded with a γ counter, and the labeling rate of ^99m^Tc-HYNIC-ADH-1 was calculated. Each experiment was repeated 4 times. 

For ^99m^Tc radiolabeling (Figure 1), SnCl_2_ was used as a reducing agent, and EDDA and Tricine were used as co-ligands. A total of 50 μL of HYNIC-ADH-1 (1 mg/mL in sterile water for injection) was added to a mixed solution of 100 μL of Tricine (100 mg/mL in Oxalic acid, pH 5.0), 100 μL EDDA (20 mg/mL in Oxalic acid, pH 5.0) and 10 μL of SnCl_2_ (3 mg/mL in 0.1 M HCl). Then, 37 MBq (1mCi) ^99m^TcO_4_^−^ was added to the solution, and the mixture was heated at 90 °C for 20 min. The radiolabeling yield was confirmed using HPLC.

### 2.5. In Vitro Stability Analysis

In vitro stability studies of ^99m^Tc-HYNIC-ADH1 were performed in FBS as well as PBS solution. A total of 50 μL of ^99m^Tc-HYNIC-ADH1 was mixed with 950 μL of FBS (or PBS), and its radioactivity was measured after being incubated at room temperature for 0.5, 1, 2, 4, 6 and 24 h, respectively.

### 2.6. Partition Coefficient Studies

Partition coefficients (Log D) of ^99m^Tc-HYNIC-ADH-1 were measured through the assessment of the distribution of radioactivity in 1-octanol and phosphate buffer in a 2 mL centrifuge tube. A 10 µL ^99m^Tc-HYNIC-ADH-1 solution was added to a vial containing 1 mL each of 1-octanol and PBS (pH 7.4). After vortexing for 10 min, the vial was centrifuged for 10 min to ensure complete separation of the layers. Then, 10 µL of each layer was pipetted into separate test tubes, and the counts were measured using a γ counter. Partition coefficients are shown as the log counts in 1-octanol vs. PBS layers (*n* = 3).

### 2.7. Fluorescence Staining, Flow Cytometry and In Vivo Imaging

#### 2.7.1. Cell Binding Study in Vitro

The PC9 and PC9GR cells were inoculated in 12-well plates (1 mL/well, 1 × 10^4^ cells per well) until the cell density reached 80%. Each well had 4% paraformaldehyde added to fix the cells and phosphate-buffered saline (PBS) was used for washing three times.

In the non-blocked group, cells were incubated for 2 h at 37 °C with 200 μL of medium per well containing different concentrations (10, 30, 50, 75 and 100 h) of Cy3-ADH-1. In the time group, cells were incubated for different times (0.5, 1, 2 and 4 h) at 37 °C with a 200 μL medium containing 50 μM of Cy3-ADH-1. Finally, 4’,6- Diamidino-2-phenylindole (DAPI) was used to stain the nuclei, and images were collected using an Olympus IX81 fluorescence microscope.

In the blocked group, PC9 and PC9GR cells were incubated for 30 min at 37 °C with a 0.5 mL medium containing 0 mM or 1 mM unlabeled ADH-1 before being incubated for 2 h with Cy3-ADH-1(100 μM, 200 μL).

#### 2.7.2. Cell Binding Rate

According to the method of cell binding study, PC9 and PC9GR cells were fixed with 4% paraformaldehyde and washed three times with PBS. In the concentration group, cells were incubated for 2 h at 37 °C with 200 μL of medium per well containing different concentrations (1, 3, 5, 8 and 10 μM) of Cy3-ADH-1. In the time group, cells were incubated for different times (0.5, 1, 2 and 4 h) at 37°C with a 200 μL medium containing 5 μM of Cy3-ADH-1. After incubation, the cells were screened and sorted using FACS (Becton, Dickinson and Company, Franklin Lakes, NJ, USA). These experiments were repeated three times, and parallel wells were included in each experiment.

The tumor-bearing mice were randomly divided into two groups (*n* = 5). In vivo fluorescence imaging was performed using a filter set with an excitation of 687 nm and emission of 712 nm on an IVIS (Caliper Lifesciences, Waltham, MA, USA), and the mice were anesthetized with 2% isoflurane before imaging. Identical illumination settings (excitation mode, exposure time and field of view) were used for acquiring all images. The non-blocked mice were injected with 100 μL Cy7-ADH-1 (1 μM) via tail vein, and subjected to optical imaging at different time points (15 min, 30 min, 1 h, 2 h, 4 h, 8 h, 24 h and 48 h). For the blocked group, ADH-1 (10 μM and 100 μL) was injected via tail vein 30 min in advance, and then the same operation was performed as that in the non-blocked group. Ellipsoidal region of interest (ROI) of equal area was drawn at bilateral tumors, and the fluorescence intensity (C/mm^2^) was quantified. At the end of imaging, the mice were sacrificed by cervical dislocation and dissected. Tumors and major organs or tissues (heart, liver, spleen, lung, kidney, stomach, intestine, muscle and bone) were collected for imaging of Cy7-ADH-1 biodistribution, and fluorescence intensity (C/mm^2^) was determined for each sample.

### 2.8. Pharmacokinetic Studies

For the pharmacokinetic studies, four C57/BL6 mice weighing 18–20 g were administrated with ^99m^Tc-HYNIC-ADH1 (0.1 mL, 0.74 MBq) by i.v. injection into the tail vein. An array of blood samples was acquired and weighed by pinching the tail with a syringe at 2, 5, 15, 20, 30, 45, 60, 90, 120 and 240 min post-injection (p.i.). Radioactivity was calculated as the % of dose per g of tissue per body weight (%ID/g) expressed over time. The radioactivity was displayed as a function of time. Pharmacokinetics was assayed using PK solver 2.0.

### 2.9. Biodistribution Studies

The radiotracer (0.1 mL, 0.74 MBq) was injected intravenously into normal C57/BL6 mice and xenograft models as previously described (*n* = 4 per group per time) via the tail vein. After different time intervals (30, 60, 120 and 240 min), the mice were sacrificed by cervical dislocation, and their blood, tumors and major organs (brain, heart, lung, liver, spleen, kidney, stomach, intestine, muscle, bone and skin) were harvested and weighed. The radioactivity of each sample was measured with a γ-counter. Values are shown as the percentage of the injected dose per gram (%ID/g), which corrected for background and decay to maintain consistency.

### 2.10. Small Animal SPECT/CT Imaging

Xenograft models were imaged using a nanoScan SPECT/CT (Mediso, Hungary) in vivo imaging system and anesthetized with 2% isoflurane before imaging. Mice in the non-blocked group (*n* = 4) were injected via tail vein with 3.7 MBq (100 μL) of radiotracer and imaged at different time points (1, 2 and 6 h). In the blocked group, 100 μL excess ADH-1 was injected through the tail vein 30 min in advance, and then the same volume and dose of imaging agent were injected for 1 h after imaging. 

Three-dimensional ROIs were delineated along the edge of the tumor using SPECT/CT analysis software. Spherical ROIs of 0.03 cm^3^ were delineated in organs or tissues (liver and muscle). The radiation intensity of each region was measured, and the tumor/non-tumor ratio was calculated.

### 2.11. Statistical Analysis

GraphPad Prism 7.0 was used for statistical analysis, and quantitative data were expressed as the mean ± SD (standard deviation) of n-independent measurements. Comparing of groups was calculated using the unpaired *t*-test. *p* < 0.05 was considered statistically significant.

## 3. Results

### 3.1. Different Influencing Factors on Labeling Yield and Radiochemistry

The results are shown in Figure 2; the optimal reaction conditions were a 20 min synthesis time of ^99m^Tc-HYNIC-ADH-1, a 10 mg dosage of Tricine was, a 50 μg dosage of HYNIC-ADH-1 and a 30 μg dosage of SnCl_2_.

According to the results of the best reaction conditions, SnCl_2_ was used as the reducing agent and EDDA and Tricine were used as co-ligands to prepare ^99m^Tc-HYNIC-ADH-1 through the indirect labeling method. The labeling rate was 97.85% ± 0.3% by ITLC-SG, and no further purification was required. Radio-HPLC was used to accurately detect the radiochemical purity of ^99m^Tc-HYNIC-ADH-1. The results (Figure 3) show that the retention time was 14.19 min and the radiochemical purity > 97%.

### 3.2. In Vitro Stability Analysis

For stability assessments, ^99m^Tc-HYNIC-ADH-1 (3.7 MBq) was incubated in PBS (pH 7.4) or FBS for 24 h. Our results reveal that the ^99m^Tc-HYNIC-ADH-1 remained stable in PBS at room temperature for 24 h, and the radiochemical purities were 99.01%, 98.81%, 98.55%, 99.02%, 98.52%, 98.01% and 99.00% (all with radiochemical purity greater than 98%) at 0.5, 1, 2, 3, 6, 12 and 24 h, respectively. The radiochemical purities were 98.51%, 98.50%, 97.97%, 98.02%, 98.81%, 98.09% and 98.09% in FBS at room temperature at 0.5, 1, 2, 3, 6, 12 and 24 h, respectively.

### 3.3. Lipophilicity (Log D) 

To determine the lipophilicity of ^99m^Tc-HYNIC-ADH-1, its partition coefficient was measured in 1-octanol and PBS (pH 7.4). The lipophilicity of ^99m^Tc-HYNIC-ADH-1 (Log D = −2.45 ± 0.10) suggested poor lipid permeability.

### 3.4. Cell Binding and Blocking Study

To investigate whether the uptake of Cy3-ADH-1 is different between the two types of cells, we performed in vitro tumor cell binding studies. As shown in Figure 4A, when the incubation time was 2 h, the binding rates of PC9GR with Cy3-ADH-1 (4.96 ± 0.74, 26.64 ± 2.78, 57.70 ± 1.56, 85.40 ± 2.97, 88.24 ± 2.46) at 1 μM, 3 μM, 5 μM, 8 Μm and 10 μM were all significantly higher than those of PC9 (1.40 ± 0.03, 13.76 ± 2.20, 25.09 ± 0.12, 46.05 ± 0.78, 61.56 ± 6.56) at corresponding concentrations (*p* < 0.05). Appendix A shows the in vitro binding of Cy3-ADH-1 with different concentrations to the two cell lines under fluorescence microscopy.

As shown in Figure 4B, when the concentration of Cy3-ADH-1 was 5.0 μM, the binding rates of PC9GR with Cy3-ADH-1 (7.96 ± 0.93, 20.45 ± 0.78, 50.05 ± 3.61, 58.00 ± 1.13) at 0.5 h, 1 h, 2 h and 4 h were all significantly higher than those of PC9 (2.92 ± 0.28, 9.46 ± 1.13, 25.09 ± 0.12, 25.78 ± 0.54) at corresponding concentrations (*p* < 0.05). The in vitro binding of Cy3-ADH-1 to the two cell lines at different times under fluorescence microscope is shown in Appendix A.

In blocking studying with an excess of ADH-1, the fluorescence intensity of Cy3-ADH-1 in PC9GR and PC9 cells at 2 h decreased after the addition of a 10-fold inhibitory amount of unlabeled ADH-1(Figure 5), and the uptake of Cy3-ADH-1 by cells could be inhibited by excessive ADH-1. These results suggest that Cy3-ADH-1 binds specifically to the N-cadherin in PC9GR and PC9 cells.

Fluorescence images of tumor-bearing mice administered with Cy7-ADH-1 probes are shown in Figure 6A. The uptake of PC9GR in the left axillary subcutaneous tissue is obvious and contrasts with the background tissue. Combined with Figure 6C, the uptake of Cy7-ADH-1 by PC9GR increased with time in the range of 15 min to 2 h, and then did not change clearly with time, and the fluorescence intensity was still at a high level at 48 h. PC9 in the right axillary subcutaneous area was not found within 15 min to 48 h. Equal area ROIs were drawn at bilateral tumors to obtain the fluorescence intensity (Figure 6C) and ratio of PC9GR and PC9 at different time points (Figure 6D). The intensity ratio of PC9GR to PC9 showed an increasing trend with time (15 min, 30 min, 1 h, 2 h, 4 h, 8 h, 24 h and 48 h were 2.19 ± 0.06, 2.26 ± 0.04, 3.01 ± 0.17, 3.42 ± 0.2, 3.31 ± 0.27, 3.91 ± 0.09, 3.80 ± 0.23 and 5.31 ± 0.4, respectively). The results of the blocked group are shown in Figure 6B. Under the same fluorescence signal range, the fluorescence intensity of PC9GR was significantly decreased 1 h p.i. (*p* < 0.05), suggesting that uptake of Cy7-ADH-1 in vivo could be inhibited by excessive ADH-1, indicating that ADH-1 had targeting ability in vivo.

In addition, the ROI covering the resected organs was plotted for in vitro assessment and quantitative assessment. As shown in Figure 6E, the PC9GR tumor in the non-blocked group still had fluorescence imaging 48 h after Cy7-ADH-1 injection, and the tumor/muscle ratio was 5.57 ± 0.32. In the blocked group, unlabeled ADH-1 reduced the uptake of Cy7-ADH-1 by PC9GR, and the T/N was 2.21 ± 0.29 (*p* < 0.01). This result is consistent with in vivo imaging. Different from in vivo imaging, the PC9 tumor in the non-blocked group showed mild uptake of Cy7-ADH-1, but there was still a difference between the PC9GR tumor and PC9 tumor (*p* < 0.05), which was consistent with the results of in vivo imaging. Excessive ADH-1 also inhibited the uptake of Cy7-ADH-1 by PC9, but less than that of PC9GR.

### 3.5. Pharmacokinetics 

The pharmacokinetic parameters obtained through the pharmacokinetic calculation program are shown in Table 1. The concentration–time curves of ^99m^Tc-HYNIC-ADH-1 in the blood of C57/BL6 mice at different times is shown in Figure 7. It is obvious that the in vivo metabolism of ^99m^Tc-HYNIC-ADH-1 in C57/BL6 mice calculated using the pharmacokinetic software PK solver is in agreement with the two-compartment model. 

### 3.6. Biodistribution Studies

In vivo biodistribution studies in normal C57/BL6 mice and xenograft-bearing nude mice are shown in Appendix A. The biological distribution of ^99m^Tc-HYNIC-ADH-1 is characterized by rapid blood clearance, with 3.31 ± 0.37 %ID/g remaining 30 min post-injection compared to 0.58 ± 0.08 %ID/g after 240 min. 

^99m^Tc-HYNIC-ADH-1 accumulates primarily in the kidneys; the radioactive concentrations at 5, 15, 30, 60, 120 and 240 min were 22.06 ± 1.15, 12.01 ± 0.73, 12.10 ± 0.47, 9.01 ± 0.76, 6.46 ± 0.30 and 6.16 ± 0.51%ID/g, respectively, indicating that the imaging agent was mainly excreted by the urinary system, which was also vividly confirmed through the time–activity derived from ROI analysis in the main organs (liver, kidney and bladder) using micro-SPECT/CT scan.

As shown in Appendix A and Figure 8A, the biodistribution in tumor-bearing mice was similar to that in normal mice. The uptake of bilateral axillary tumor tissues was significantly higher than that in muscle, and the uptake of PC9 gradually decreased with the delay of time (30 min: 0.99 ± 0.15 %ID/g, 240 min: 0.64 ± 0.16 %ID/g), while PC9GR increased step by step in the range of 30 min to 2 h, reached a peak (1.32 ± 0.08 %ID/g), then slightly decreased.

The ratios of tumor/non-tumor (muscle, liver and kidney) at each time point are listed in Table 2. In tumor/muscle, the PC9GR/muscle ratio was higher than that of PC9 at each time point, and both of them peaked at 2 h. The difference between PC9GR/muscle and PC9/muscle at 30 min, 1 h and 2 h was statistically significant ( *p*< 0.05, Figure 8B).

### 3.7. Small Animal SPECT/CT Imaging

^99m^Tc-HYNIC-ADH-1 micro-SPECT/CT images are shown in Figure 9A. Tumor (PC9GR and PC9) uptake was evident at the bilateral of the mice where NSCLC cells were inoculated, peaked at 1 h post-injection and then declined over time. The overall uptake of the PC9GR tumor was higher than the PC9 tumor. 

The ROI analysis of the ratio of radioactivity intensity at tumor sites to the radioactivity intensity at adjacent muscles (T/N) showed that the T/N of the uptake of ^99m^Tc-HYNIC-ADH-1 by the PC9GR tumor was 7.73 ± 0.54 at 1 h, which was higher than the T/N of the PC9 tumor (3.66 ± 0.78). 

In the blocked group, as shown in Figure 9C, the unlabeled ADH-1 significantly reduced ^99m^Tc-HYNIC-ADH-1 PC9GR tumor uptake (T/N = 3.69 ± 0.50, *p *< 0.01), confirming the specificity of the labeled probe.

## 4. Discussion

N-cadherin is considered one of the mesenchymal markers of EMT [12]. In NSCLC, studies have found that high N-cadherin expression is closely related to gefitinib resistance and lung cancer brain metastasis [14,15]. Given that N-cadherin is involved in cancer development and cancer dissemination, and the important role of drug resistance, research on its inhibitors is important [16]. ADH-1 is the most-studied N-cadherin inhibitor, and in vitro and clinical experiments have proven that apoptosis can be induced in many kinds of tumor cells to improve drug sensitivity [18,20,21]. In phase I clinical trials, ADH-1 was proven to be a promising anticancer drug, with acceptable toxicity, validating N-cadherin as a potential target for treatment of tumor drug resistance [19,24].

In this study, the optical molecular probe Cy3-ADH-1 was synthesized for cell binding experiments in vitro, which showed that both PC9 and PC9GR cells could bind to Cy3-ADH-1 (Appendix A), and the binding rate of drug-resistant PC9GR cells and Cy3-ADH-1 was significantly higher than that of drug-sensitive PC9 cells (Figure 4). To detect the specificity of Cy3-ADH-1, unlabeled ADH-1 was used for competitive inhibition of cellular uptake. The results (Figure 5) showed that the uptake of Cy3-ADH-1 by PC9 and PC9GR cells was significantly reduced by excessive unlabeled ADH-1, which was in line with the law of competitive binding. To further prove the targeting and specificity of ADH-1 in vivo, we synthesized a Cy7-ADH-1 molecular probe to perform fluorescence dynamic imaging in tumor-bearing mice. The results (Figure 6) showed that the uptake of Cy7-ADH-1 in PC9GR tumors reached a maximum value at 2 h, and the signal was slowly washed away with time. High contrast with background tissue could be observed during the whole process for tumors derived from resistant PC9GR tumors but not for those derived from PC9 tumors. During the imaging process, the fluorescence intensity ratio of PC9GR to PC9 was greater than 2:1, which was consistent with the results of the in vitro cell binding experiment. The ratio of PC9GR and PC9 also showed an overall increasing trend with the extension of time and peaked at 48 h, with a ratio of 5.31 ± 0.4. In subsequent tissue and organ imaging, PC9GR tumors still showed significant uptake of Cy7-ADH-1 and a high tumor–tissue ratio (Figure 6). In addition, in vivo competition experiments showed that excessive amounts of unlabeled ADH-1 significantly reduced the uptake of Cy7-ADH-1 by PC9GR tumors. Our research demonstrated the high targeting ability and specificity of ADH-1 to PC9GR cells. However, fluorescence imaging has limited depth and is mainly used for superficial tissue detection in external or endoscopic surgery, and radiometric imaging is more commonly used in whole-body imaging [25]. 

Radiological imaging is a relatively mature technique. Multi-imaging increases the accuracy of anatomical localization results for radionuclide imaging. In the clinic, ^18^F-fluorodeoxyglucose positron emission tomography (^18^F-FDG PET) is widely used because of the abnormal glucose metabolism in most tumors, and has become an important tool for malignant tumor staging and efficacy evaluation [26]. SPECT is a traditional radionuclide imaging technology that is simpler and more economical than PET. Due to the different imaging principles, the resolution is slightly worse than that of PET. Small animal SPECT/CT is a fusion imaging instrument for mice or rats that is used for early studies of new imaging agents before they are translated into clinical practice, and can predict clinical imaging results to a certain extent [27]. Small animal SPECT has higher resolution and sensitivity than conventional SPECT [28]. In addition, the radionuclide commonly used in SPECT, ^99m^Tc, is a gamma-emitter with an energy of 141 KeV and a half-life of 6.02 h [29]. It is easy to obtain at low cost and is the most commonly used radionuclide for imaging at present. 

Small molecular peptides have the advantages of small molecular weight, easy synthesis, high specificity and quick blood clearance. Compared with antibodies, peptides have no immunogenicity and are highly specific for targeting tumors. Compared with ligands, peptides have more functional groups and can provide more sites of access, making them ideal materials for molecular probes [30]. As a cyclic peptide, ADH-1 has significantly improved antidegradation ability, affinity and biological activity compared with linear peptides [31,32]. Radiolabeling of peptides is generally stable and tolerant of structural modifications as long as the binding basic sequence is retained [33]. We used an indirect labeling method to prepare the radioactive probe. The mixture was heated at 90 °C for 20 min with SnCl_2_ as the reducing agent and EDDA and tricine as coligands. The labeling process was simple and rapid. The labeling rate was more than 97% by ITLC-SG and HPLC, and no further purification was needed. ^99m^Tc-HYNIC-ADH-1 is highly stable in PBS and FBS and the radiochemical purity is more than 98% at 24 h, which meets the requirements for further study of ^99m^Tc-HYNIC-ADH-1 in vivo. 

The lipid water partition coefficient (LogD) is the ratio of the compound concentration at equilibrium between the hydrophobic phase (1-octanol) and the hydrophilic phase. The smaller the lipid water partition coefficient, the higher the solubility of the compound in the aqueous phase and the lower the solubility in the lipid phase. We calculated the LogD of ^99m^Tc-HYNIC-ADH-1 was −2.45 ± 0.10, indicating that it had strong water solubility and poor lipid permeability, suggesting that the molecular probe was mainly excreted through the urinary system. We confirmed this hypothesis with subsequent biodistribution and SPECT imaging.

Biodistribution studies (Appendix A) showed that ^99m^Tc-HYNIC-ADH-1 was characterized by rapid blood clearance, with 3.31 ± 0.37 %ID/g remaining 30 min p.i. compared to 0.58 ± 0.08 %ID/g after 240 min. ^99m^Tc-HYNIC-ADH-1 accumulated primarily in the kidneys, indicating that the imaging agent was mainly excreted by the urinary system, which was not conducive to the observation and analysis of posterior peritoneum and pelvic tissues. Nevertheless, rapid clearance is desirable for diagnostic tracers, as it reduces tissue exposure to radiation and background noise. In addition, there was no obvious distribution of ^99m^Tc-HYNIC-ADH-1 in the brain, indicating that it could not cross the blood–brain barrier.

Biodistribution in PC9GR and PC9 tumor-bearing nude mice (Figure 8) and SPECT/CT imaging (Figure 9A) showed significant uptake by engrafted tumors. The semiquantitative analysis revealed that the T/NT of the PC9GR tumors was significantly higher than that of the PC9 tumors at 1 h after intravenous injection of ^99m^Tc-HYNIC-ADH-1 (*p* < 0.01). This optimal imaging time allows sufficient time for preparation, as is the case for ^18^F-FDG. In the blocked group, a low level of inhomogeneous radiation distribution was observed in bilateral tumor tissues, and the T/NT of PC9GR was significantly reduced (*p* < 0.01). Our results preliminarily indicate that gefitinib-resistant NSCLC cells can specifically take up ^99m^Tc-HYNIC-ADH-1 and the uptake capacity is higher than that of drug-sensitive NSCLC cells. This molecular probe is promising for the imaging of drug-resistant NSCLC.

## 5. Conclusions

Based on the fact that EMT is one of the mechanisms of drug resistance in tumor cells, our team conducted a preliminary study on the molecular probe of tumor drug resistance by using the principle of specific binding of ADH-1 to N-cadherin, one of the mesenchymal landmark proteins. The ^99m^Tc-HYNIC-ADH-1 molecular probe has a short synthesis time, does not require purification, has a high labeling rate and radiochemical purity, has good stability, is characterized by rapid blood clearance and is mainly excreted through the urinary system. Preliminary results showed that the ^99m^Tc-HYNIC-ADH-1 molecular probe could be used for imaging of drug-resistant NSCLC, providing a noninvasive method to dynamically observe whether NSCLC patients develop drug resistance during treatment. Fluorescently labeled ADH-1 can be used to evaluate tissue samples and guide surgical resection.

## Figures and Tables

**Figure 1 cancers-15-00755-f001:**
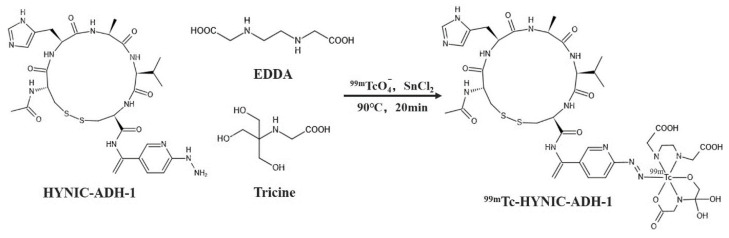
Labeling reaction of ^99m^Tc-HYNIC-ADH-1.

**Figure 2 cancers-15-00755-f002:**
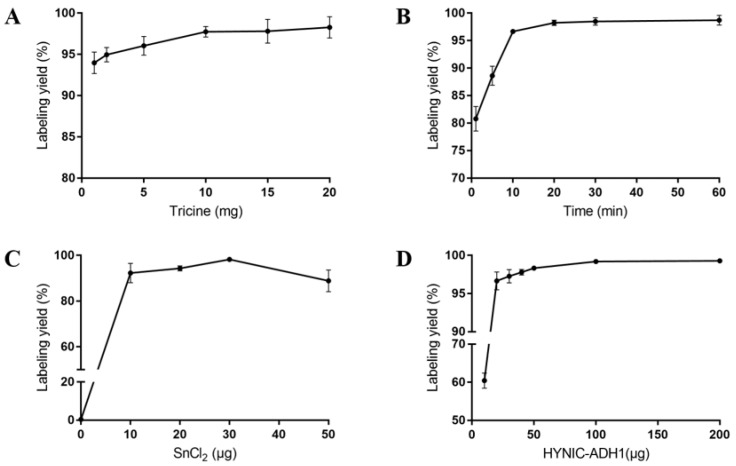
The optimum reaction conditions of ^99m^Tc labeled HYNIC-ADH-1. (**A**) The effect of reaction time on labeling yield (%). (**B**) The effect of tricine dosage on labeling yield (%). (**C**) The effect of SnCl_2_ dosage on labeling yield (%). (**D**) The effect of HYNIC-ADH-1 dosage on labeling yield (%). All data are expressed as mean ± SD (*n* = 4).

**Figure 3 cancers-15-00755-f003:**
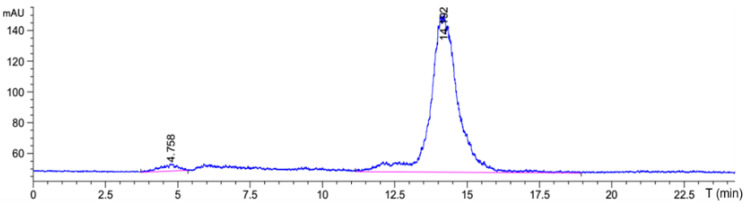
Radio-synthesis and HPLC radio-chromatogram of ^99m^Tc-HYNIC-ADH-1.

**Figure 4 cancers-15-00755-f004:**
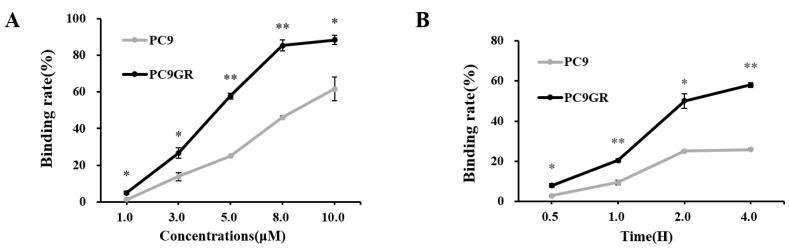
Cell binding rates of Cy3-ADH-1 in PC9GR and PC9. (A) The binding rates increased with the increase in Cy3-ADH-1 concentration at 2 h. (B) The binding rates increased with the increase in incubation time at 5.0 μM. (* *p* < 0.05, ** *p* < 0.01).

**Figure 5 cancers-15-00755-f005:**
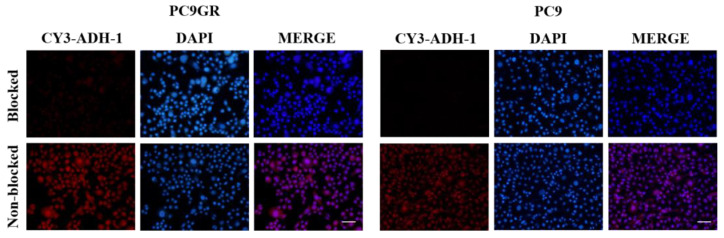
Effect of excessive ADH-1 on uptake of Cy3-ADH-1 in PC9GR and PC9 cells through a fluorescence microscope (100×). The cells of the blocked group had 1 mM unlabeled ADH-1 added 30 min in advance. Scale bar, 100 μm.

**Figure 6 cancers-15-00755-f006:**
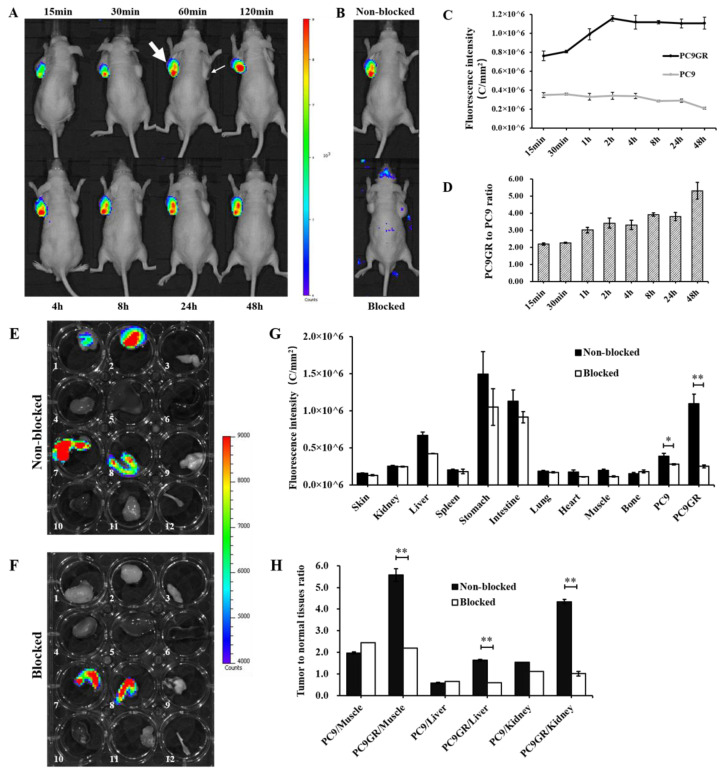
(**A**) Dynamic fluorescence imaging in vivo of Cy7-ADH-1 after intravenous administration (thick arrow: PC9GR tumor, thin arrow: PC9 tumor). (**B**) Fluorescence imaging of mice in non-blocked group and blocked group 1 h p.i. of Cy7-ADH-1. (**C**) Fluorescence intensity of PC9GR and PC9 at different time points after Cy7-ADH-1 injection in tumor-bearing mice. (**D**) PC9GR to PC9 ratios at different time points. In vitro imaging of tumors and major organs or tissues 48 h p.i. of cy7-ADH-1 in non-blocked group (**E**) and blocked group (**F**): 1 PC9 tumor, 2 PC9GR tumor, 3 skin, 4 kidney, 5 liver, 6 spleen, 7 stomach, 8 intestine, 9 lung, 10 heart, 11 muscle, 12 bone. Biodistribution of fluorescence intensity ex vivo of tumors and major tissues in non-blocked and blocked group (**G**) and ratio of tumor to normal tissues (**H**) based on ROI analysis. (* *p* < 0.05, ** *p* < 0.01).

**Figure 7 cancers-15-00755-f007:**
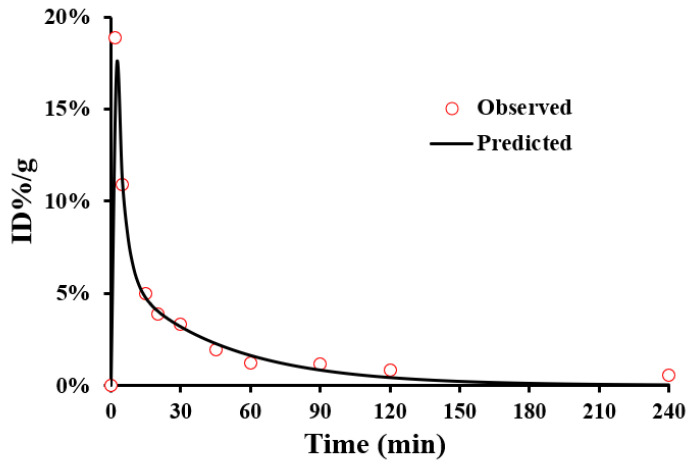
The blood drug concentration (%ID/g)-time curve for ^99m^Tc-HYNIC-ADH-1 in C57/BL6 mice until 240 min after injection.

**Figure 8 cancers-15-00755-f008:**
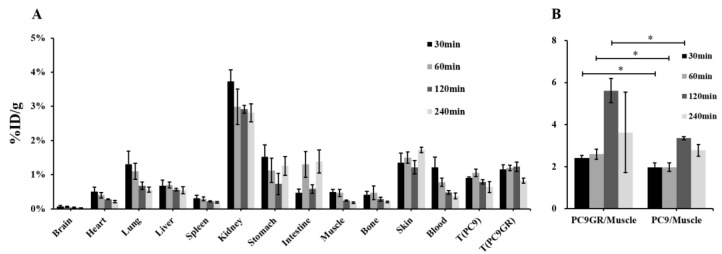
(**A**) PC9 and PC9GR tumor-bearing BALB/c nude mice biodistribution studies of ^99m^Tc-HYNIC-ADH-1 at 30, 60, 120 and 240 min p.i. Values are expressed as %ID/g. (**B**) Tumor to muscle ratio of ^99m^Tc-HYNIC-ADH-1 in tumor-bearing mice (* *p* < 0.05).

**Figure 9 cancers-15-00755-f009:**
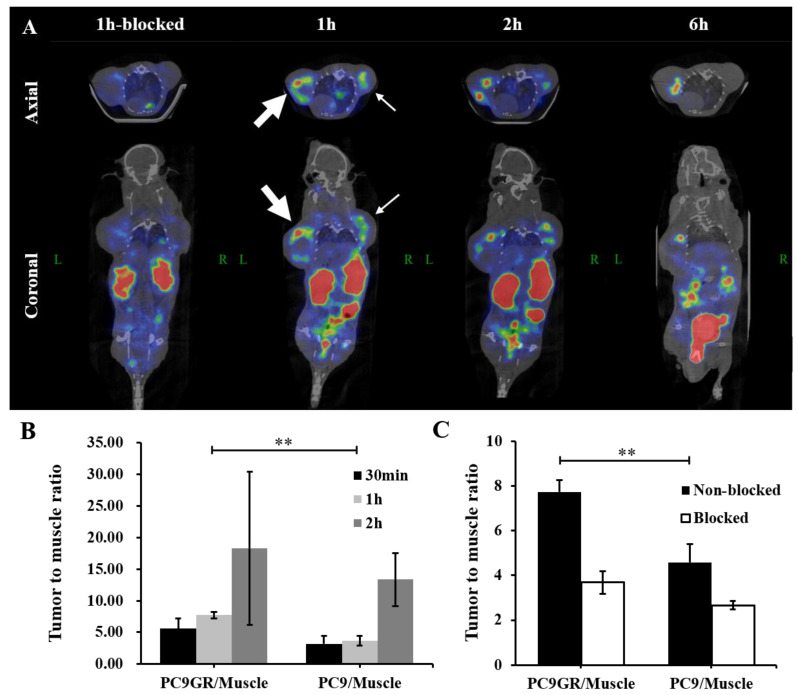
(**A**) Dynamic imaging of ^99m^Tc-HYNIC-ADH-1 micro-SPECT/CT in mouse model at different times (thick arrow: PC9GR tumor, thin arrow: PC9 tumor). (**B**) Tumor (PC9GR and PC9) to muscle ratio at 30 min, 1 h and 2 h based on ROI analysis. Since no radioactivity was found in the muscle at 6 h, the data of the 6 h group were not included. (**C**) Tumor (PC9GR and PC9) to muscle ratio of non-blocked group mice and blocked group mice at 1 h based on ROI analysis. (** *p* < 0.01).

**Table 1 cancers-15-00755-t001:** Pharmacokinetic parameters of ^99m^Tc-HYNIC-ADH-1 in C57/BL6 mice (AUC, area under the curve; CL, clearance).

Parameters	Value
T_1/2α_ (min)	2.3925
T_1/2β_ (min)	31.0256
V_1_ (L/kg)	68.753
CL (L/min/kg)	336.2721
K_10_ (min^−1^)	0.0815
K_12_ (min^−1^)	0.1511
K_21_ (min^−1^)	0.0794
AUC_(0–t)_ (μg/L·min)	0.0593
AUC_(0–∞)_ (μg/L·min)	0.0595

**Table 2 cancers-15-00755-t002:** Tumor/non-tumor ratio of ^99m^Tc-HYNIC-ADH-1 in tumor-bearing mice (x ± SD).

T/NT	Time			
	30 min	1 h	2 h	4 h
PC9GR/Muscle	2.41 ± 0.12	2.64 ± 0.47	5.19 ± 1.89	3.62 ± 1.92
PC9/Muscle	1.97 ± 0.21	2.59 ± 0.34	3.36 ± 0.08	2.78 ± 0.28
PC9GR/Liver	1.64 ± 0.24	1.58 ± 0.12	2.14 ± 0.69	1.36 ± 0.69
PC9/Liver	1.36 ± 0.15	1.58 ± 0.06	1.41 ± 0.21	1.01 ± 0.31
PC9GR/Kidney	0.42 ± 0.10	0.27 ± 0.04	0.49 ± 0.20	0.26 ± 0.13
PC9/Kidney	0.27 ± 0.06	0.27 ± 0.03	0.26 ± 0.06	0.25 ± 0.02

## Data Availability

The original contributions presented in the study are included in the article/Appendix A.

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
