# Peer review of "Development of 99mTc-Hynic-Adh-1 Molecular Probe Specifically Targeting N-Cadherin and Its Preliminary Experimental Study in Monitoring Drug Resistance of Non-Small-Cell Lung Cancer"

_cancers, 2023, doi:10.3390/cancers15030755_

Round 1

Reviewer 1 Report

Ye et al. develop a novel molecular imaging agent based on the N-cadherin inhibitor that could be used for imaging drug-resistant NSCLC, providing a non-invasive method for dynamically observing whether tumor resistance occurs during treatment. This is an interesting study. It will be very useful for monitoring drug resistance if it succeeded.

1. ADH-1 (N-Ac-CHAVC-NH2) has been shown to induce apoptosis of various tumor cells, inhibit tumor cell migration, and improve drug sensitivity in both in vitro and preclinical experiments. Does it affect the viability of parent and resistant cell lines? Does the probe affect cell viability? Is there any possibility that the cell binding rate difference between parent and resistant cell lines is caused by the different viability?

2. For fluorescence images, borders must be used to clearly distinguish images. Each part of the figure should be crisp and clear. There should show a scale bar.

3. Should show some evidence that the cell line is Gefitinib-resistant.

4. Is there any difference in N-cadherin level between parent and resistance cell lines?

5. Does this probe binding to other cadherins?

Reviewer 2 Report

The authors present a compelling manuscript that can be improved in a few aspects prior to publication.

1. Please cite all supplementary figures in the text. also check supplementary figures for Chinese language.

2. Necessary edits to introduction:

line 45 - add "EGFR" prior to gene mutations

line 46 - many or most would be more accurate than "some"

lines 54-62 - this section is lacking in relevant citations from the field.

line 71 - do you mean the "role" of N-cadherin?

line 72 - add "of N-cadherin" after inhibitors

line 75 - replace "It" with "ADH-1"

line 77 - "improve drug sensitivity" is vague

line 86 - authors must state what HYNIC is, and spell out the acronym

3. Methods, line 219 - please specify what type of t-test was used.

4. line 251, authors need to specify what the 2 cell types are.

5. Figure 4A - legend and text need to include incubation time.

6. Figure 4 - additional statistical detail is required. Please add asterix to indicate degree of significance at at each observation point (e.g. *=P<0.05, **=p<0.01 etc.)

7. Figures 5-8 - these images are not possible to interpret as presented. I believe the data supports the conclusions, but this reviewer has to spend a lot of time trying to discern differences between images.

8. Line 281 - I believe this figure call-out is incorrect.

9. Figure 8 - authors need to include images from mice with PC9 tumors either here or in the supplement.

10. line 300 - i do not understand the reference to in vitro assessment? do the authors mean ex vivo?

11. Table 2 is almost impossible to interpret. Can the authors please find some way to graphically demonstrate the key findings and move the table to the supplement.

12. Figure 11 - can the authors please explain the dramatic increase in variability observed in the PCRGR/Muscle ratio at 240 minutes.

13. Figure 12 - please label the tumor types.  Also, the tumors appear to be of different sizes - this must be accounted for when looking at absorbtion ratios.

14. Discussion and conclusions require some editing for clarity (e.g. first sentence of line 403)

Round 2

Reviewer 2 Report

Changes are acceptable.